# Hippocampal Neurogenesis Is Enhanced in Adult Tau Deficient Mice

**DOI:** 10.3390/cells9010210

**Published:** 2020-01-14

**Authors:** Marangelie Criado-Marrero, Jonathan J. Sabbagh, Margaret R. Jones, Dale Chaput, Chad A. Dickey, Laura J. Blair

**Affiliations:** 1Department of Molecular Medicine, USF Health Byrd Institute, University of South Florida, Tampa, FL 33613, USA; marangelie@usf.edu (M.C.-M.); jonathan.sabbagh@nih.gov (J.J.S.); margaret.r.jones@vanderbilt.edu (M.R.J.); cdickey@health.usf.edu (C.A.D.); 2Department of Cell Biology, Microbiology and Molecular Biology, University of South Florida, Tampa, FL 33620, USA; chaput@mail.usf.edu

**Keywords:** tauopathies, depression, hippocampal neurogenesis, stress, glucocorticoid receptor, Alzheimer’s disease

## Abstract

Tau dysfunction is common in several neurodegenerative diseases including Alzheimer’s disease (AD) and frontotemporal dementia (FTD). Affective symptoms have often been associated with aberrant tau pathology and are commonly comorbid in patients with tauopathies, indicating a connection between tau functioning and mechanisms of depression. The current study investigated depression-like behavior in *Mapt^−/−^* mice, which contain a targeted deletion of the gene coding for tau. We show that 6-month *Mapt^−/−^* mice are resistant to depressive behaviors, as evidenced by decreased immobility time in the forced swim and tail suspension tests, as well as increased escape behavior in a learned helplessness task. Since depression has also been linked to deficient adult neurogenesis, we measured neurogenesis in the hippocampal dentate gyrus and subventricular zone using 5-bromo-2-deoxyuridine (BrdU) labeling. We found that neurogenesis is increased in the dentate gyrus of 14-month-old *Mapt^−/−^* brains compared to wild type, providing a potential mechanism for their behavioral phenotypes. In addition to the hippocampus, an upregulation of proteins involved in neurogenesis was observed in the frontal cortex and amygdala of the *Mapt^−/−^* mice using proteomic mass spectrometry. All together, these findings suggest that tau may have a role in the depressive symptoms observed in many neurodegenerative diseases and identify tau as a potential molecular target for treating depression.

## 1. Introduction

The microtubule-associated protein tau accumulates in the brain in several neurodegenerative diseases, including Alzheimer’s disease (AD) [1]. This tau accumulation leads to neuronal dysfunction, pervasive synaptic loss, and cognitive deficits [2,3]. Likewise, tau pathology strongly correlates with neuron death and severity of dementia while affecting brain regions in the limbic system [4,5], suggesting that tau is critically important in the progressive development of behavioral symptoms observed in AD. Symptoms of depression, in particular, are commonly co-morbid with AD and other neurodegenerative diseases [6], leading to more rapid decline and higher mortality [7,8]. However, it is not known if this is a direct consequence of neuropathological changes or a response to the psychological burden of dementia. Moreover, while recent evidence has demonstrated that depression can predict subsequent risk for developing neurodegenerative diseases [9,10,11,12], it is unclear if this represents a prodromal stage of the disease or merely a risk factor [13].

Depression-like behavior in rodent models of AD has not been extensively studied, despite the prevalence of depression co-morbidity in patients [6]. However, it has been demonstrated that aged 3xTg-AD mice, which carry insertions of mutant tau, human amyloid precursor protein (hAPP), and presenilin, displayed increased immobility time in the forced swim and tail suspension tests, measures of depression-like behavior in rodents during middle (10-month-old) [14] and older (18-month-old) [15] ages. Similarly, aged (13–15 month-old) but not young (5–7 month-old) hAPP mutant mice exhibited depression-like behavior that was accompanied by decreased neurogenesis [16]. Interestingly, decreased neurogenesis and hippocampal-dependent memory deficits were observed in both ages’ groups independent of sex factor [16]. Other studies have shown that mice overexpressing β-amyloid (Aβ) displayed a depressive-like behavior [17,18], suggesting Aβ may play a role in the depressive symptoms observed in AD. Considering that 3xTg-AD and hAPP transgenic mice present a reduction of hippocampal neurogenesis [19,20,21], it has been theorized that this reduction in neurons may contribute to increased susceptibility to depression. However, whether tau plays a direct role in this phenotype and directly contributes to neurogenesis in adult mice remains unknown. 

The current study investigated depression-like behavior in *Mapt^−/−^* mice, which contain a targeted deletion of the gene coding for tau [22]. Here, we confirm a prior finding that tau deficient mice are resistant to stress and depressive behaviors [23]. Although this work only observed a resiliency to depression after stress exposure and only measured these effects in male mice, our work builds on this by testing both male and female *Mapt^−/−^* mice under normal conditions, as evidenced by decreased immobility time in the tail suspension tests, as well as increased escape behavior in a learned helplessness task. Additionally, we found that hippocampal neurogenesis was increased in 14-month-old *Mapt^−/−^* mice. Using mass spectrometry, we identified over 2500 proteins in the hippocampus, amygdala and frontal cortex which structures perform key functions in emotion and cognitive processes. We detect alterations in proteins involved in neuronal development and signaling. Since AD patients present dysregulation of cortisol levels and high levels of glucocorticoids promotes tau accumulation and upregulation of aberrant tau phosphorylated species in wild type middle aged rats, we assessed if the *Mapt^−/−^* behavioral phenotype was due to changes in the glucocorticoid receptor (GR) signaling [24,25]. However, proteomic analysis as well as in vitro and in vivo assessments showed that protection from depression was independent of GR signaling. Overall, our findings demonstrate that tau may have a central role in manifesting depressive symptoms observed in many neurodegenerative diseases and identifies tau as a new molecular target for treating depression in AD.

## 2. Materials and Methods

### 2.1. Animals

*Mapt^−/−^* mice (B6.129X1-MAPTtm1Hnd/J) were obtained from Jackson Laboratory (Bar Harbor, ME, USA) and genotyped as previously described [22]. Wild type C57BL/6J mice (Jackson Laboratory, Bar Harbor, ME, USA) were used as controls in all experiments. Mice were group housed under a 12-h light-dark cycle (lights on at 06:00) and permitted ad libitum access to food and water. Behavioral studies were all conducted on wild type (WT) and tau knockout (*Mapt^−/−^*) littermates at 6 months of age, who were then sacrificed at 14 months of age. A total of 15–20 mice were used per group and genotype for behavioral testing. Each group contained a balanced number of males and females. All animal studies were approved by the University of South Florida Institutional Animal Care and Use Committee and carried out in accordance with the National Institutes of Health (NIH) guidelines for the care and use of laboratory animals.

### 2.2. Tail Suspension Test

The tail suspension test (TST) consisted of suspending each mouse by the tail for one 6-min session, as previously described [26]. Test was recorded using the Any-MAZE software (Stoelting Co., Wood Dale, IL, USA) and the time immobile was measured by a blinded experimenter.

### 2.3. Forced Swim Test

The forced swim test (FST) was conducted as previously described [26]. Mice were placed into a clear glass cylinder 23 cm tall with a 15 cm diameter, filled with room temperature water to a depth of 13 cm. Mice were exposed and recorded for one 6-min session using Any-MAZE software (Stoelting Co., Wood Dale, IL, USA). A trained experimenter blinded to genotype measured the total immobile time for each animal.

### 2.4. Learned Helplessness Paradigm

The learned helplessness, LH, paradigm was performed according to previously established protocols [27]. Briefly, mice were placed in a two-way shuttle box (Coulbourn Instruments, Whitehall, PA, USA) and given 60 s to habituate to the arena. Animals were then subjected to 360 mild, inescapable foot shocks (0.15 mA) of 1–3 s duration with a 1–15 s inter-shock interval. This training session was repeated 24 h later followed by the assessment of learned helplessness behavior 24 h later. The assessment was performed in the same shuttle box with 30 trials of a 10 s shock followed by a 30 s interval. The shock was terminated when the mouse traversed to the other side of the box (an “escape”) or at the termination of the shock with no attempt to escape (a “failure”). The number of escapes and failures and latency to escape were recorded for each animal using the infrared beams at the base of the compartments and Graphic State software (Coulbourn Instruments, https://www.coulbourn.com/).

### 2.5. Open Field 

Mice were placed individually in an empty square-shaped environment and allowed to explore for 5 min, as previously described [28]. Distance travelled and time spent in the center was tracked by video using ANY-maze software.

### 2.6. Nociceptive Behavior

Nociception was measured using the hot plate test as previously described [29]. Mice were placed on a plate set to 52 °C and latency to withdraw the paw and/or tail was recorded by a trained experimenter blind to genotype.

### 2.7. Corticosterone Enzyme-Linked Immunosorbent Assay (ELISA)

A total of non-stressed (10 mice per genotype) and stressed (9–10 mice per genotype) mice were used for corticosterone (CORT) measurements. Each group contained a balanced number of males (~5) and females (~5). Blood from wild type and *Mapt^−/−^* mice was collected via the submandibular vein one hour following the start of the light cycle (basal) and 30 min following a 10 min tube restraint (stressed) [30]. Serum was separated from whole blood using microtainer serum separator tubes (BD, Franklin Lakes, NJ, USA) and centrifuged for 15 min at 1300× *g*. Levels of serum CORT were quantified using a CORT ELISA kit (Enzo Life Sciences, Farmingdale, NY, USA), as recommended by the manufacturer. 

### 2.8. Dexamethasone Suppression Test

To asses hypothalamic-pituitary-adrenal (HPA) axis function, *Mapt^−/−^* and wild type mice received an intraperitoneal injection with dexamethasone (0.05 mg/kg). Blood was collected in the morning at 7:30 am (baseline sample, pre-DEX injection) and in the afternoon at 2:30 pm (6-h later, post-DEX injection). A total of wild type (*n* = 10) and *Mapt^−/−^* (*n* = 12) mice were used for this test. Levels of serum CORT were quantified using a CORT ELISA kit (Enzo Life Sciences, Farmingdale, NY, USA), as described above.

### 2.9. Bromodeoxyuridine Administration

5-Bromodeoxyuridine (BrdU) was prepared fresh in a saline solution at a concentration of 10 mg/mL. Mice were injected intraperitoneally with 50 mg/kg body weight once per day over 5 consecutive days. To allow full incorporation of BrdU into newly divided cells, mice were euthanatized 3 weeks after the final injection at 14- months of age.

### 2.10. Immunohistochemistry

Immunohistochemistry was conducted as previously described [31]. Briefly, 14- months old mice were overdosed with pentobarbital and transcardially perfused with 0.9% saline solution. Brains were removed and hemispheres were separated. Hemi-brains were placed in 4% paraformaldehyde overnight prior to cryoprotection in increasing sucrose gradients (10%, 20% and 30% each day). Hemi-brains were sectioned horizontally on a freezing microtome at a thickness of 25 µm and stored in phosphate-buffered saline containing 0.02% NaN_3_ at 4 °C until staining. Six sections containing the hippocampus were selected for staining. Immunofluorescent staining was completed using anti-BrdU (1:200; AbD Serotec) and anti-NeuN (1:100; Millipore) primary antibodies with Alexa Fluor 594 and 633 secondaries (1:1000; Invitrogen). Sections were then mounted in glass slides and allowed to dry overnight, followed by cover-slipping the next day. Slides were scanned into digital images using a Zeiss AxioScan.Z1 slide scanner. Positive staining was determined visually based on morphology and co-localization of BrdU with NeuN and counted by hand for both the dentate gyrus and subventricular zones.

### 2.11. Mass Spectrometry 

Hippocampus (HPC), amygdala (AMG), and frontal cortex (FCX) of 14-month-old *Mapt^−/−^* and wild type mice (*n* = 3 mice per genotype) were homogenized in 5 µL per mg brain structure weight of lysis buffer (100 mM Tris-HCl, 100 mM DTT, 4% SDS w/v, pH 7.6 and 1:100 protease and phosphatase inhibitors). Homogenates were lysed at 95 °C and sonicated, followed by Pierce 660 concentration assay with ionic detergent compatibility reagent. Samples were then digested with trypsin using filter-aided (FASP), desalted and dried on a speed-vac prior to mass spectrum analysis. 

Peptides were separated on a 50 cm C18 reverse-phased UHPLC column (Thermo Scientific) using an EZnano1000 HPLC (Thermo Scientific) with a 120 min gradient and analyzed on a hybrid quadrupole-Orbitrap instrument (Q Exactive Plus, Thermo Scientific). Full MS survey scans were acquired at 70,000 resolution and using data-dependent acquisition (DDA) the top 10 most abundant ions were selected for MS/MS analysis.

Raw data files were processed in MaxQuant (www.maxquant.org) and searched against the UniprotKB Mus Musculus protein sequence database. Search parameters included constant modification of cysteine by carbamidomethylation and the variable modification, methionine oxidation, and phosphorylation of serine, threonine, and tyrosine. Proteins were identified using the filtering criteria of 1% protein and peptide false discovery rate.

Label-free quantitation analysis was performed using Perseus, software developed for the analysis of omics data (Cox & Mann, 2016). Briefly, intensities were Log2-transformed, and then filtered to include proteins containing at least 60% valid values (reported LFQ intensities) in at least one experimental group. Finally, the missing values in the filtered dataset are replaced using the imputation function in Perseus with default parameters (Cox & Mann, 2016). Statistical analyses is carried out using the filtered and imputed protein groups file.

Statistical analysis was performed using Scaffold software and sorted by *p*-value and fold change (*Mapt^−/−^* over WT mice). Proteins with *p*-values less than 0.05 were then uploaded to the Ingenuity Pathway Analysis (IPA) software (QIAGEN Inc., Germantown, MD, USA). Canonical pathways, diseases and functions, protein and gene networks were evaluated for each brain region using IPA.

### 2.12. Cell Culture, Transfection, and Luciferase Activity Assay

M17 cells were transfected with MAPT and control siRNA for a total of 72 h using siLentFect (Bio-Rad, Hercules, CA, USA) and were subsequently transfected with a GRE-luciferase reporter, a GR cDNA plasmid and pRL-CMV using Lipofectamine 2000 (Invitrogen, Waltham, MA, USA) for 48 h, as previously described [32]. For the last 4 h cells were exposed to 50 nm Dexamethasone or DMSO. 10% charcoal-stripped FBS was used throughout (Life Technologies, Carlsbad, CA, USA). Cells were lysed and subjected to Dual-Luciferase reporter assay kit (Promega, Madison, WI, USA) as previously described [32,33]. A parallel experiment was lysed in MPER and 30 µg of lysate was evaluated by Western blotting, as we have done previously [32], using Tau 12 (kind gift from Nicholas Kanaan) and Actin (Sigma Aldrich, St. Louis, MO, USA) antibodies.

## 3. Results

### 3.1. Mapt^−/−^ Mice Display a Depression-Resistant Phenotype

We investigated the impact of tau deletion on depressive-like behaviors by exposing *Mapt^−/−^* mice to various behavioral tests. We found that, compared to WT, *Mapt^−/−^* mice are resistant to depression-like behavior, as evidenced by a significant decreased immobility in the tail suspension (Figure 1A; *p* < 0.001) and forced swim (Figure 1B; *p* < 0.05) tests, as well as decreased failures and latency to escape in the learned helplessness task (Figure 1C,D; *p* < 0.05). Using the open field test, we then confirmed that neither locomotion (Figure 1E) nor anxiety levels (Figure 1F) were affected in *Mapt^−/−^* mice, confirming that tau deletion selectively impacted depressive-like behaviors. Then we measured nociception to ensure that the decreased LH behavior was not due to impaired pain response and therefore less fear/pain from the footshock in the LH test. *Mapt^−/−^* mice exhibited similar latency in paw withdrawal using the hot plate test as compared to WT mice (Figure 1G) demonstrating normal response to pain and that the mechanisms involved in pain sensitization may be independent of tau. Interestingly, sex contributed to the observed depressive phenotypes without affecting anxiety, locomotion or pain response (Appendix A). Here, females showed greater resiliency to depressive-like symptoms when compared to males (Figure 2A–D), which does align with depression being more prevalent in women [34].

### 3.2. Resiliency to Depression Is Independent of the HPA Axis Function

Since the function of the hypothalamic-pituitary-adrenal (HPA) axis is known to be dysregulated in patients with AD and/or depression [35,36], we evaluated whether GR signaling is affected by tau in in vitro and in vivo experiments. First, we transfected a human neuroblastoma cell line, M17 cells, with control or MAPT siRNAs followed by confirmation of downregulation of tau protein (Figure 3A). In a second experiment, we utilized a GR reporter luciferase assay to measure GR activity in M17 cells. We observed an upregulation in GR activity after 4 h of dexamethasone exposure (Figure 3B; *p* < 0.05).

We then tested if the HPA axis was altered after DEX exposure as being a possible mechanism underlying the protective phenotype against depression in *Mapt^−/−^* mice. However, we did not observe changes in basal or stress-induced serum CORT levels between *Mapt^−/−^* and control mice (Figure 4A). These results were validated using the dexamethasone suppression test, which is commonly used to examine HPA axis function in patients with depression [37]. In both control and experimental groups, CORT levels increased similarly following dexamethasone injection (Figure 4B). Our results also indicate that sex did not impact CORT levels after stress or DEX induction (Appendix A). These findings suggest that despite that tau deletion can affect GR activity in cells, this effect does not translate to in vivo observations since HPA signaling was unaffected in *Mapt*^−/−^ mice. This indicates that GR signaling is not responsible for conferring protection from depression in these mice.

### 3.3. Tau Deletion Increases Neurogenesis in the Dentate Gyrus and Subventricular Zone

Since depression has been strongly linked to decreased neurogenesis [38] and tau deletion prevents neuronal loss [39], we speculated that this process was altered in *Mapt^−/−^* mice. To measure neurogenesis, we counted positive proliferating cells using BrdU labeling. We found that neurogenesis was increased in the dentate gyrus (Figure 5A,B) and subventricular zone (Figure 5C) in *Mapt^−/−^* brains, providing a potential mechanism for their protective behavioral phenotype towards depression. Increased neurogenesis (*p* < 0.05) was also evident when measuring DCX+ cells (Appendix A) and this upregulation was independent of sex (Figure 5D,E). These results demonstrate that tau plays an important role in neurogenesis and its dysregulation can impact cell viability in mental and neurodegenerative diseases.

### 3.4. Processes Involved in Neuronal Development Are Significantly Upregulated in Mapt^−/−^ Mice

Besides the hippocampus, other brain structures in the cortico-limbic system such as the amygdala and frontal cortex are altered in depression [40,41]. Using mass spectrometry, we performed a proteomic analysis of these three brain areas to examine if neuronal processes were affected in 14-month-old *Mapt^−/−^* mice. Over 2500 proteins were identified in each brain region after filtering and imputation [42]. Proteins with a Welch’s *t*-test *p*-value < 0.05 were uploaded to Ingenuity Pathways Analysis (IPA) for bioinformatic analysis. Compared to control mice, *Mapt^−/−^* mice had increased expression of proteins involved in processes like neuronal development and proliferation, neurite formation, growth and branching (Figure 6A). Since neuronal branching is essential for establishing neuronal circuits and promoting communication, the increased branching indicates more axonal and dendritic projections that may stimulate synapses and communication among neurons. This upregulation was observed in all brain areas demonstrating that tau deletion promotes neuronal growth and development throughout the limbic system. In addition, proteomic analysis showed that the formation of neurites, neuritogenesis, was the most influenced process in the hippocampus of tau deficient mice. The fold change of top proteins was assessed (Figure 6B).

## 4. Discussion

Although there is a high prevalence of depression in patients with Alzheimer’s disease, the etiology and mechanisms involved is still under investigation. In this study, we demonstrate that tau levels can regulate stress-induced behavior and neuronal health using a tau knockout animal model. Our data clearly show that *Mapt^−/−^* mice were protected from depressive-like symptoms as well as physiological without overt effects on the HPA axis function. We also found that, in 14-month-old mice, deletion of tau enhances hippocampal neurogenesis and proteins involved in neuronal development in corticolimbic structures like the hippocampus, amygdala, and frontal cortex.

There are a plethora of preclinical and clinical studies suggesting a strong association between chronic stress exposure and depression as well as Alzheimer’s disease [10,15,43]. Evidence suggests that stress contributes to the etiology of these diseases by negatively impacting the HPA axis adaptive response, altering neuronal function, inducing both genetic and epigenetic changes among other pathological changes in the brain [44]. Consistent with previous studies [23,45], we report that 6-month-old *Mapt^−/−^* mice were protected from stress, as evidenced by unchanged serum CORT levels after acute restraint stress or dexamethasone injection. In line with this, the proteomic analysis did not reveal significant changes in proteins associated with glucocorticoid signaling. Altogether, these findings suggest that tau is not necessary to mediate HPA axis or brain adaptive response in these disorders. But, we know that stress and high CORT promotes tau accumulation and upregulation of aberrant tau phosphorylated species in wild type middle aged rats [24]. Therefore, we cannot exclude the possibility of tau indirectly acting as a positive feedback modulator potentiating CORT-induced dendritic atrophy, neuronal death, and cognitive impairments in AD patients exposed to stress independent of the HPA axis [46,47]. This may explain how stress increases susceptibility to develop both diseases. Additional studies are needed to investigate if long-term stress plays a role in promoting comorbidity for depression and AD since patients with dual diagnosis present similar CORT-induced alterations like abnormal brain activity, lower neurometabolites (like N-acetylaspartate), and reduced cortical thickness when compared to AD without depression [48,49,50].

In addition to the behavioral and physiological response, *Mapt^−/−^* mice are protected from negative stress-induced effects in neurons [51,52]. At a similar age to our cohort, 3XTg-AD, hTau, and hAPP transgenic mice have shown a reduction in neurogenesis [19,20,21,53]. Dioli et al. reported increased neurogenesis in young *Mapt^−/−^* male mice [54]. Here we extend this finding by including females in our study and showing that increased neurogenesis remained until later stages during middle adult stage. Our data also demonstrated that tau depletion promoted neurogenesis in 14-month-old *Mapt^−/−^* mice. Considering that tau has an essential role in maintaining microtubule stability and neuronal morphogenesis, it is possible that there is a compensatory effect from other microtubule-associated proteins promoting neurogenesis. Specifically, MAP1A expression is increased in primary hippocampal cultures as well as hippocampal and brain homogenates from tau deficient mice. Thus, it may compensate when tau is lacking [22,55,56]. Moreover, our proteomic analysis confirms that not only tau is essential for hippocampal neurogenesis, but also it modulates important processes like neuronal development and proliferation in structures like amygdala and frontal cortex. This is important because these brain structures play an essential synergistic role in psychiatric and aging disorders as well are influenced by early life experiences and stress [57].

Moreover, a depressive phenotype has been reported in neurogenesis-deficient mice [38]. Accordingly, reduced neurogenesis has not only been associated in AD [58], but also this pathological hallmark is evident in depression [59] consistent with smaller hippocampi [60,61,62,63]. Increasing neurogenesis and proteins associated to neuronal development may be the mechanism underlying the protection against depression in *Mapt^−/−^* mice. Here, we used proteomics analysis to identify potential pathways influencing neurogenesis in tau deficient mice. Different from what we were expecting, GR signaling was not affected in these mice. Our data showed that hippocampal neuritogenesis was the most impacted neuronal process, and this was accompanied by the highest fold change in Pak3. Pak3 is a serine-threonine kinase commonly known as a downstream effector of CDC42 and RAC1, two members of the Rho Family GTPases. It mediates processes like cytoskeletal remodeling, synaptic plasticity and neurogenesis [64,65,66,67]. Interestingly, two independent studies showed that Pak3 expression is significantly reduced in the hippocampus and frontal cortex of postmortem brains from patients with depression and AD [68]. Also, Pak3 interacts with APP when overexpressed or with familial AD mutants of APP mediating neuronal apoptosis [66]. Our findings suggest that tau also regulates Pak3 signaling in the hippocampus. Additionally, the protein with the highest fold change in the frontal cortex, Rhob, is also a member of the Rho Family GTPases (Appendix A). Although more research is needed to determine a direct interaction of tau and Rho Family GTPases as well as their effect on neurogenesis, these proteins may be viable targets to treat depression in AD patients.

Previous studies using hTau and 3xTg-AD mice suggest that tau dysregulation may be implicated in the development of depressive symptoms in AD [14,15,69]. Therefore, combined with our study, this suggests that by reducing tau in the brain we may promote the growth and development of new neurons, which may prevent or delay brain atrophy associated with dementia and depression [70]. One possible treatment to restore neurogenesis is by using the GR antagonist, mifepristone, which has shown to be effective in ameliorating depressive symptoms while normalizing neuronal growth and reducing levels of β-amyloid (Aβ) and tau phosphorylated species [71,72,73,74]. Another possibility is treating patients with antidepressants. Although there still controversy about their use in AD patients, preclinical studies have demonstrated that reduced neurogenesis can be restored by antidepressants. For example, antidepressant drugs (like fluoxetine) and interventions (like repeated electroconvulsive shock) promotes precursor cell proliferation, neurogenesis, and neurite arborization [75,76]. Commonly used antidepressants such as selective serotonin reuptake inhibitors (SSRI) are designed to extend the availability of neurotransmitters in the synaptic cleft. Since mice deficient in the serotonin neurotransmitter showed reduced depression-like behavior and chronic administration of the SSRI fluoxetine lead to increased hippocampal neurogenesis [77], this may be a feasible medication to restore neurogenesis in adults. In preclinical and clinical studies a single treatment of the antidepressant ketamine has been sufficient to relieve depressive symptoms in adult patients and C57BL/6 mice previously exposed to chronic stress [78,79]. In humans, ketamine is known to relief symptoms in a very short period of time, but more investigation is needed to ameliorate or prevent associated side effects [79]. In mice, the treatment with ketamine reduced aggregation of hyperphosphorylated tau protein in the hippocampus of these mice after stress exposure. A more holistic intervention may also be possible including: environmental enrichment and exercise, which are known to enhance neurogenesis and stimulate the production of neurotrophic factors while improving cognition and mood disorder symptoms [80,81,82]. Each of these interventions may provide neuroprotection through the induction of transcription factors (e.g., cAMP-responsive element binding protein, CREB), neuronal growth factors (e.g., BDNF), proteins (e.g., PSD95), and receptors (e.g., AMPA) involved in synaptic plasticity. They also inhibit proinflammatory markers (e.g., NF-KB) and reuptake of neurotransmitters, prolonging their availability in the synapse for receptor binding, improving cognition and mood [83,84,85,86]. This is important because together with neurotransmitters, these factors directly impact synaptic plasticity affecting cognitive and emotional processes. In this line, research findings from adult rats demonstrated that induction of long-term potentiation can enhance neurogenesis contributing to hippocampal-dependent memory [87]. Moreover, 3xTg-AD and Tau KO mice have reduced NMDA-dependent LTP and deficits in hippocampal long-term depression, respectively [88,89]. It is possible that by stimulating neurogenesis we could improve the synaptic deficits reported in AD and depression. Additionally, genetic variations in genes like FKBP5, which is stress-induced, can affect antidepressant treatment efficacy [90]. Although more research is needed, this study demonstrated that high expression of FKBP5 correlated with low BDNF expression, which may affect essential neural processes such as neurogenesis and synaptic plasticity in the brain [90]. In addition, high FKBP5 levels have also been shown to be found in the AD brain and can preserve neurotoxic tau [91].

An interesting observation in the present study was that female *Mapt^−/−^* mice showed greater protective phenotype to depression than males. We also found that *Mapt^−/−^* females also showed higher neurogenesis in the subventricular zone when compared to males. Our results add to multiple studies suggesting that sex may contribute to differences in the prevalence and development of AD and depression. For example, researchers using an AD mouse model found that females exhibit higher number of amyloid plaques and neurofibrillary tangles, markers of neuroinflammation, and spatial cognitive deficits [92]. In a separate study, young male mice that overexpressed β-amyloid plaques showed decreased serotonin and dopamine levels as compared to females with the same genotype [93]. It is good to point out that dysregulation of both the serotonin and dopamine signaling is implicated in depression and Alzheimer’s disease [94,95,96], thus including both sexes provides essential information about the disease. This sexual dimorphism should not be surprising since research in Alzheimer’s and depression have demonstrated sex differences during brain development, susceptibility to stress, HPA and hippocampal regulation, as well as treatment response in preclinical and clinical settings [97,98,99].

Also, the exclusion of a sex group in previous studies makes it harder to compare results and make assertive conclusions on how tau and stress affects behavior differently in both sexes. It is important to note that a prior report did not find a difference in depression-like behavior in *Mapt^−/−^* [23]. Another difference from prior studies is that we did not observe changes in nociception in the *Mapt^−/−^* mice [100]. However, these studies included only males and used a different mouse strain (C57BL/6), which may explain the difference in our results. These differences in results highlight the importance of including both sexes in preclinical and clinical studies. Lack of representation from both sexes can provide unassertive and incomplete information limiting the progress in understanding the etiology of AD and depression, as well as drug development and treatment options. Considering that sex differences may influence treatment efficacy and responses to behavioral therapies, here we describe behavioral and molecular distinctions in both sexes to better understand the effect of using tau as a pharmacological target.

## 5. Conclusions

Overall, we demonstrate that young *Mapt^−/−^* mice, from both sexes, have a protective phenotype from depression. We showed that neuronal development is promoted not only in the hippocampus but also the frontal cortex and amygdala in adult *Mapt^−/−^* mice. Enhanced neuronal development in these brain areas indicates that tau may have a role regulating cognitive and emotional processes affecting depression. Although behavioral and molecular changes were independent of GR signaling, counter to our original hypothesis, the proteomic analysis revealed other molecular pathways regulated by tau, including Rho Family GTPases. This information is valuable to better understand the effects of targeting tau and the main molecular pathways regulated in tauopathies.

## Figures and Tables

**Figure 1 cells-09-00210-f001:**
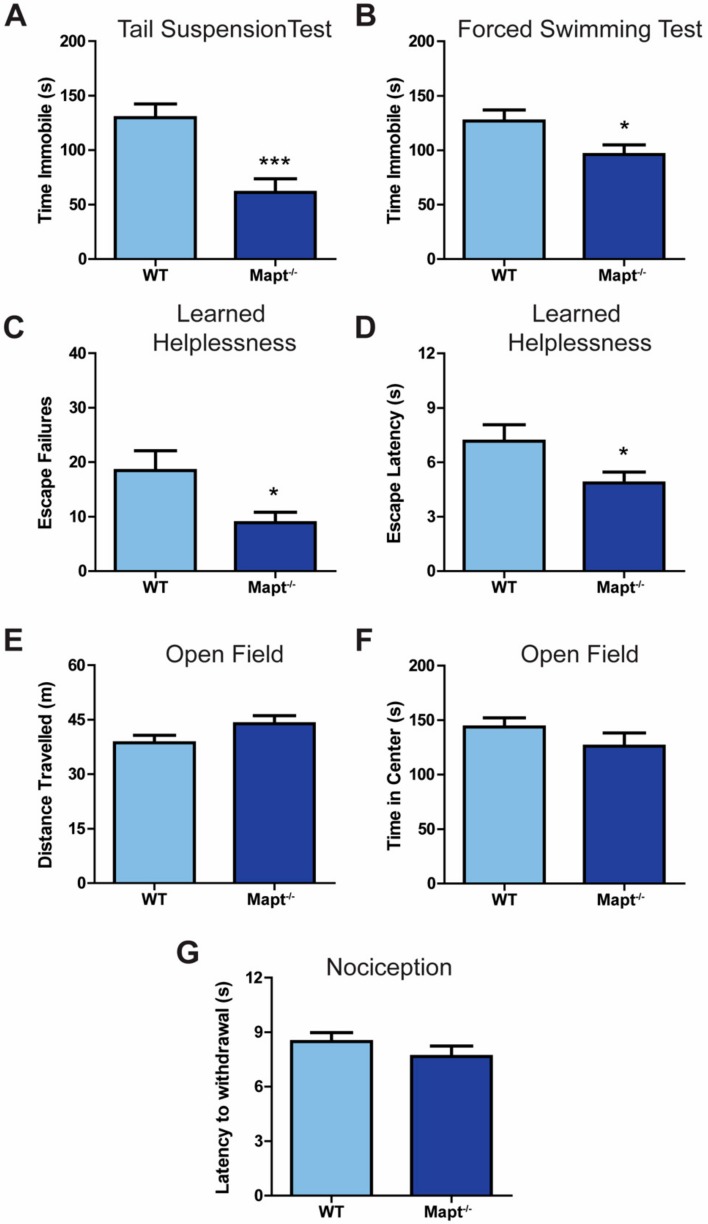
Tau ablation increased resiliency to depressive-like symptoms. Immobility time was measured for *Mapt^−/−^* and WT mice using the (**A**) tail suspension and (**B**) forced swim tasks. In the learned helplessness paradigm, (**C**) escape failures and (**D**) escape times were measured. (**E**) Total distance traveled and (**F**) time spent in the center in the open field task was measured for *Mapt^−/−^* and WT mice. (**G**) A hot-plate was used to measure nociception in *Mapt^−/−^* and WT mice. Data are represented as standard error of the mean (SEM) and analyzed by Student’s *t*-test. WT, wild type. A total number of 15–20 animals were used per genotype. Significant results were considered when * *p* < 0.05, *** *p* < 0.001.

**Figure 2 cells-09-00210-f002:**
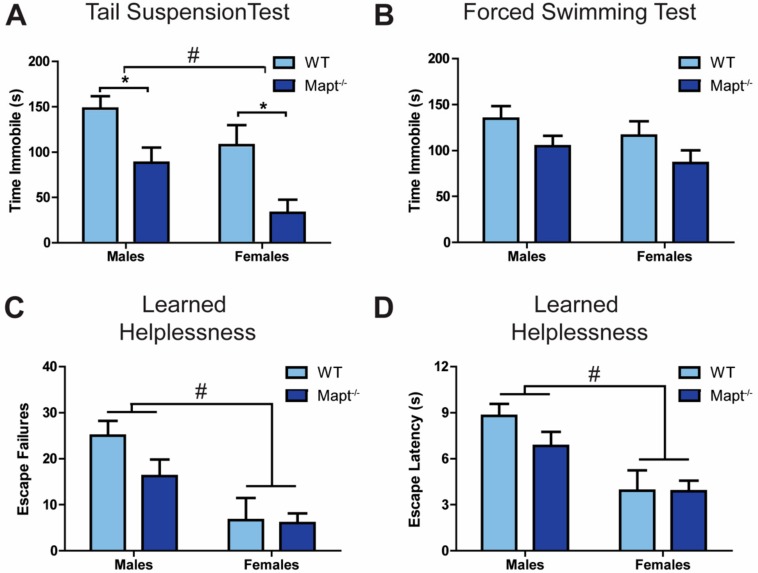
Depressive-like behavior is more pronounced in female mice. Data from Figure 1 broken down by sex for *Mapt*^−/−^ and WT mice in the (**A**) tail suspension and (**B**) forced swim tasks, and (**C**,**D**) the learned helplessness paradigm. Data are represented as standard error of the mean (SEM) and analyzed by two-way ANOVA followed by Bonferroni posthoc test. WT, wild type. Significant results by genotype * *p* < 0.05 or by sex # *p* < 0.05.

**Figure 3 cells-09-00210-f003:**
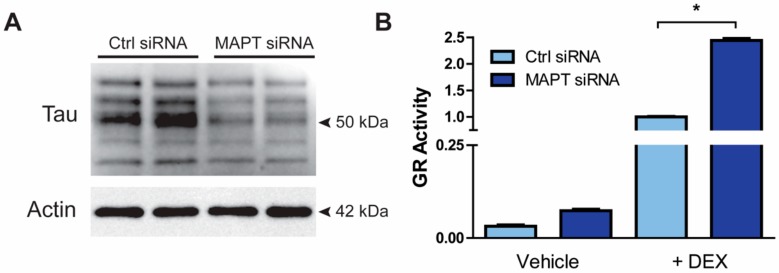
Tau knockdown increased GR activity in vitro. (**A**) M17 cells were incubated with either control or MAPT siRNAs for 72 hrs. Following transfection, cells lysates were collected and reduction of tau levels was confirmed by Western blot. (**B**) After MAPT and control siRNA transfection, M17 cells were cotransfected with a GRE-luciferase reporter and a GR cDNA plasmid for 2 days prior to treatment with vehicle (DMSO) (*n* =3) or 50 nM DEX (*n* =3) for 4 h. Experiment was run in triplicate. GR activity was measured using a GRE reporter luciferase assay. Data are represented as standard error of the mean (SEM) and analyzed by Student’s *t*-test. Significant results were considered when * *p* < 0.05. DEX, dexamethasone; GR, glucocorticoid receptor; MAPT siRNA, short interfering RNA targeting MAPT.

**Figure 4 cells-09-00210-f004:**
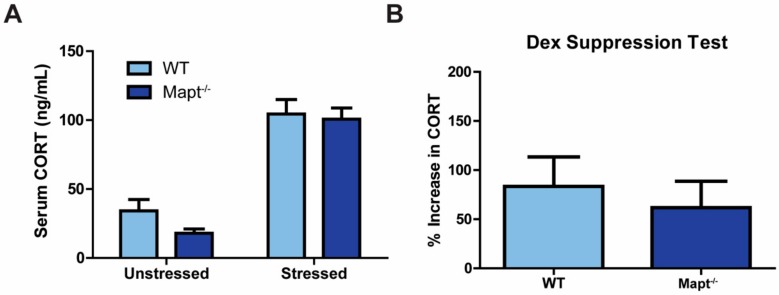
HPA axis function was unaltered in tau knockout mice. (**A**) Serum CORT levels were measured by ELISA in *Mapt^−/−^* and WT mice in unstressed and stressed conditions (*n* = 10 per group). (**B**) CORT induction following dexamethasone injection was measured in serum from *Mapt^−/−^* and WT mice (WT = 10, *Mapt^−/−^* = 12). Data are represented as standard error of the mean (SEM) and analyzed by two-way ANOVA (non-stressed vs. stressed) or Student’s *t*-test (dexamethasone suppression test). CORT, corticosterone; Dex, dexamethasone; WT, wild type.

**Figure 5 cells-09-00210-f005:**
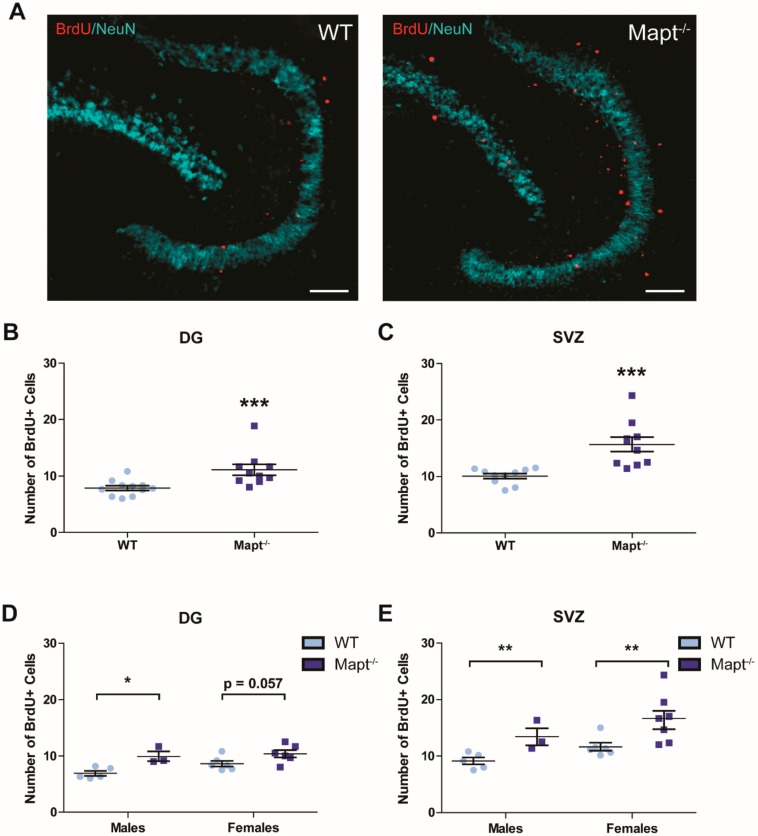
Upregulation of neurogenesis occurs in the dentate gyrus and subventricular zone of *Mapt^−/−^* mice. Brains from 14- months-old WT and *Mapt^−/−^* mice were collected and stained with anti-BrdU (proliferating cell marker) and anti-NeuN (neuronal nuclei marker). (**A**) A representative 10× image of BrdU+/NeuN staining in *Mapt^−/−^* and WT tissue is shown. Scale bar represents 100 µm. Quantification of BrdU+ staining was performed in the (**B**) DG and (**C**) SVZ. Quantification of BrdU+ staining by sex is also shown in the (**D**) DG and (**E**) SVZ brain areas. Data are represented as standard error of the mean (SEM) and analyzed by two-way ANOVA (neurogenesis by sex) and Student’s *t*-test (WT vs. *Mapt^−/−^*). Significant results are shown as * *p* < 0.05, ** *p* < 0.01, *** *p* < 0.001. DG, dentate gyrus; SVZ, subventricular zone; WT, wild type; BrdU+, 5-bromo-2-deoxyuridine positive cells.

**Figure 6 cells-09-00210-f006:**
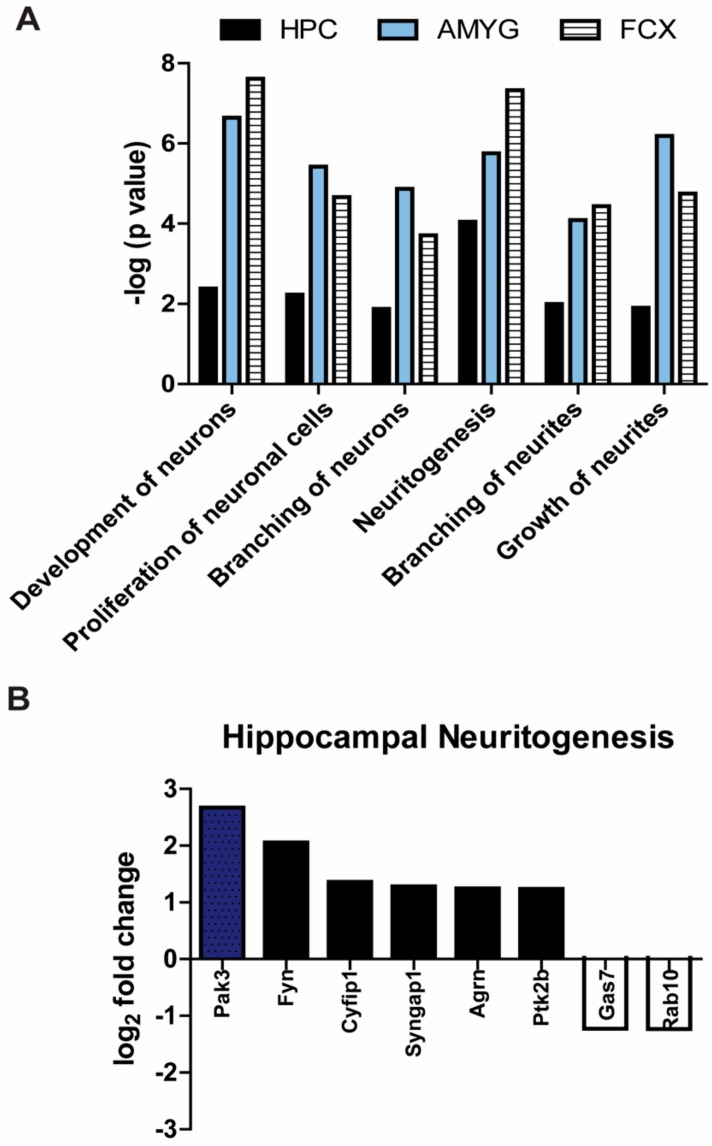
Processes involved in neurogenesis are significantly upregulated in adult *Mapt^−/−^* mice. (**A**) Neural tissue homogenates from three different brain regions Hippocampus (HPC); amygdala (AMYG); frontal cortex (FCX) of 14-month-old *Mapt^−/−^* and WT mice were analyzed by proteomic mass spectrometry and then by Ingenuity Pathway Analysis software. We examined top processes involved in neurogenesis. (**B**) Our proteomics data showed that neuritogenesis was greatly increased in the hippocampus, so we selected the top proteins altered in this process. Fold change of proteins in the frontal cortex and amygdala can be found in Appendix A. Data are displayed as difference in *Mapt^−/−^* mice compared to WT (*n* = 3 per genotype).

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
