# Peer review of "Hippocampal Neurogenesis Is Enhanced in Adult Tau Deficient Mice"

_cells, 2020, doi:10.3390/cells9010210_

Round 1

Reviewer 1 Report

The authors have adequately addressed all critiques.

Author Response

Reviewer 1: The authors have adequately addressed all critiques.

Response: We greatly appreciate all comments and constructive criticism provided by the reviewer to improve the quality of our manuscript.

Reviewer 2 Report

Tau is a microtubule-associated protein, which accumulates in the brain of tauopathy neurodegenerative disease patients. In this manuscript, Criado-Marrero et al. first confirm that Mapt-/- (tau gene KO) mice showed higher resistance to depression-like behaviors, and then investigated possible underlying mechanisms. Authors found increased neurogenesis in Mapt-/- mouse and increase in proteins related to neurogenesis. They also report sex differences in depressive behaviors. It is suggested that depression accelerates the development of neurodegenerative diseases. This article includes important information on tau’s function in the depression. The studies are organized theoretically, the data seems evident, and the manuscript is written well. I have minor comments only as described below.

Minor comments:

Figure 2A; There are several bands, all of which are reduced by Mapt siRNA. Are all of them tau? If not, indicate tau band(s). If they are, mention it. Legend of Fig. 4A, 400 micro meter, but not micro molar. Explain what the branching of neurons is. Line 295, Pak3 is a Ser/Thr kinase but not a member of Rho GTPase. Pak3 is a downstream effector of Cdc42 and Rac, members of Rho GTPases. Sex differences in the depression behavior may be included in the main text but not shown as the supplementary, as authors discuss it using the relatively large space in the Discussion.

Author Response

Reviewer 2: Tau is a microtubule-associated protein, which accumulates in the brain of tauopathy neurodegenerative disease patients. In this manuscript, Criado-Marrero et al. first confirm that Mapt-/- (tau gene KO) mice showed higher resistance to depression-like behaviors, and then investigated possible underlying mechanisms. Authors found increased neurogenesis in Mapt-/- mouse and increase in proteins related to neurogenesis. They also report sex differences in depressive behaviors. It is suggested that depression accelerates the development of neurodegenerative diseases. This article includes important information on tau’s function in the depression. The studies are organized theoretically, the data seems evident, and the manuscript is written well. I have minor comments only as described below.

Response:

Figure 2A; There are several bands, all of which are reduced by Mapt siRNA. Are all of them tau? If not, indicate tau band(s). If they are, mention it. Thanks for the comment. Tau often shows multiple bands particularly in the range between 48-67 kDa due to the 6 isoforms and many post-translational modifications. We have now updated the blot image in now Figure 3A to have an indicated molecular weight marker for reference. Legend of Fig. 4A, 400 micro meter, but not micro molar. We apologize for not catching this. We have corrected the measurement unit in the legend of now Figure 5. Explain what the branching of neurons is. Thank you for pointing this out. In the results (section 3.4), we have included more information about the importance of this mechanism in the nervous system.

Since neuronal branching is essential for establishing neuronal circuits and promoting communication, the increased branching indicates more axonal and dendritic projections that may stimulate synapses and communication among neurons.”

Line 295, Pak3 is a Ser/Thr kinase but not a member of Rho GTPase. Pak3 is a downstream effector of Cdc42 and Rac, members of Rho GTPases. Thank you for your comment. We have corrected this in the text.

“Pak3 is a serine-threonine kinase commonly known as a downstream effector of CDC42 and RAC1, two members of the Rho Family GTPases.”

Sex differences in the depression behavior may be included in the main text but not shown as the supplementary, as authors discuss it using the relatively large space in the Discussion. Thank you for this suggestion. We have incorporated the sex differences into the main text as new Figure 2.

Reviewer 3 Report

Tau protein has a significant function as a microtubule stabilizer.

Besides, Mapt-/- mice model has shown changes in neuronal maturation and morphology, observing delayed maturation of hippocampal neurons in primary hippocampal cultures ( Dawson, H.N., Ferreira, J. Cell Sci. 114 (Pt 6), 1179–1187.
Interestingly, here the authors demonstrated that mice Mapt-/- are resistant to depressant symptoms as compared to wild type. Also, mice have shown augmented neurogenesis in the dentate gyrus and subventricular zone. However, since tau is crucial for microtubule maintenance, the authors might indicate whether the density level of microtubules in the dentate gyrus and subventricular area remain the same or not.
If not, who might be compensating this facts?  

Author Response

Reviewer 3Besides, Mapt-/- mice model has shown changes in neuronal maturation and morphology, observing delayed maturation of hippocampal neurons in primary hippocampal cultures (Dawson, H.N., Ferreira, J. Cell Sci. 114 (Pt 6), 1179–1187. Interestingly, here the authors demonstrated that mice Mapt-/- are resistant to depressant symptoms as compared to wild type. Also, mice have shown augmented neurogenesis in the dentate gyrus and subventricular zone. However, since tau is crucial for microtubule maintenance, the authors might indicate whether the density level of microtubules in the dentate gyrus and subventricular area remain the same or not. If not, who might be compensating these facts? 

Response:

Thanks for pointing this out. As you indicate, tau is known to be essential for microtubule-binding and tubulin-polymerizing activity stabilizing microtubules. Since it is not possible to test microtubule density in our samples because of the limited time requested for resubmission and resources available, we have included a possible explanation for promoting neurogenesis in tau deficient mice in the current draft. But we do agree that this is worth more attention in follow-up studies where a more careful evaluation of the microtubule density in various subregions can be performed. Our suggestion is based on previous data showing increased levels of other members of the microtubule-associate protein family. We have added the following statement in the discussion:

“Considering that tau has an essential role in maintaining microtubule stability and neuronal morphogenesis, it is possible that there is a compensatory effect from other microtubule-associated proteins promoting neurogenesis. Specifically, MAP1A expression is increased in primary hippocampal cultures as well hippocampal and brain homogenates from tau deficient mice, thus it may compensate when tau is lacking [22,55,56]”.